# The Sound of APALM Clapping: Faster Nonsmooth Nonconvex Optimization with Stochastic Asynchronous PALM

**Damek Davis and Madeleine Udell**
Cornell University
{dsd95,mru8}@cornell.edu

**Brent Edmunds**
University of California, Los Angeles
brent.edmunds@math.ucla.edu

## Abstract

We introduce the Stochastic Asynchronous Proximal Alternating Linearized Minimization (SAPALM) method, a block coordinate stochastic proximal-gradient method for solving nonconvex, nonsmooth optimization problems. SAPALM is the first asynchronous parallel optimization method that provably converges on a large class of nonconvex, nonsmooth problems. We prove that SAPALM matches the best known rates of convergence — among synchronous or asynchronous methods — on this problem class. We provide upper bounds on the number of workers for which we can expect to see a linear speedup, which match the best bounds known for less complex problems, and show that in practice SAPALM achieves this linear speedup. We demonstrate state-of-the-art performance on several matrix factorization problems.

## 1 Introduction

Parallel optimization algorithms often feature synchronization steps: all processors wait for the last to finish before moving on to the next major iteration. Unfortunately, the distribution of finish times is heavy tailed. Hence as the number of processors increases, most processors waste most of their time waiting. A natural solution is to remove any synchronization steps: instead, allow each idle processor to update the global state of the algorithm and continue, ignoring read and write conflicts whenever they occur. Occasionally one processor will erase the work of another; the hope is that the gain from allowing processors to work at their own paces offsets the loss from a sloppy division of labor.

These *asynchronous parallel* optimization methods can work quite well in practice, but it is difficult to tune their parameters: lock-free code is notoriously hard to debug. For these problems, there is nothing as practical as a good theory, which might explain how to set these parameters so as to guarantee convergence.

In this paper, we propose a theoretical framework guaranteeing convergence of a class of asynchronous algorithms for problems of the form

$$\underset{(x_1,\ldots,x_m)\in\mathcal{H}_1\times\ldots\times\mathcal{H}_m}{\text{minimize}} f(x_1,\ldots,x_m) + \sum_{j=1}^{m} r_j(x_j), \tag{1}$$

where $f$ is a continuously differentiable ($C^1$) function with an $L$-Lipschitz gradient, each $r_j$ is a lower semicontinuous (not necessarily convex or differentiable) function, and the sets $\mathcal{H}_j$ are Euclidean spaces (i.e., $\mathcal{H}_j = \mathbb{R}^{n_j}$ for some $n_j \in \mathbb{N}$). This problem class includes many (convex and nonconvex) signal recovery problems, matrix factorization problems, and, more generally, any generalized low rank model [20]. Following terminology from these domains, we view $f$ as a *loss function* and each $r_j$ as a *regularizer*. For example, $f$ might encode the misfit between the observations and the model, while the regularizers $r_j$ encode structural constraints on the model such as sparsity or nonnegativity.

Many synchronous parallel algorithms have been proposed to solve (1), including stochastic proximal-gradient and block coordinate descent methods [22, 3]. Our asynchronous variants build on these synchronous methods, and in particular on proximal alternating linearized minimization (PALM) [3]. These asynchronous variants depend on the same parameters as the synchronous methods, such as a step size parameter, but also new ones, such as the maximum allowable delay. Our contribution here is to provide a convergence theory to guide the choice of those parameters within our control (such as the stepsize) in light of those out of our control (such as the maximum delay) to ensure convergence at the rate guaranteed by theory. We call this algorithm the Stochastic Asynchronous Proximal Alternating Linearized Minimization method, or SAPALM for short.

Lock-free optimization is not a new idea. Many of the first theoretical results for such algorithms appear in the textbook [2], written over a generation ago. But within the last few years, asynchronous stochastic gradient and block coordinate methods have become newly popular, and enthusiasm in practice has been matched by progress in theory. Guaranteed convergence for these algorithms has been established for convex problems; see, for example, [13, 15, 16, 12, 11, 4, 1].

Asynchrony has also been used to speed up algorithms for nonconvex optimization, in particular, for learning deep neural networks [6] and completing low-rank matrices [23]. In contrast to the convex case, the existing asynchronous convergence theory for nonconvex problems is limited to the following four scenarios: stochastic gradient methods for smooth unconstrained problems [19, 10]; block coordinate methods for smooth problems with separable, convex constraints [18]; block coordinate methods for the general problem (1) [5]; and deterministic distributed proximal-gradient methods for smooth nonconvex loss functions with a single nonsmooth, convex regularizer [9]. A general block-coordinate stochastic gradient method with nonsmooth, nonconvex regularizers is still missing from the theory. We aim to fill this gap.

**Contributions.**  We introduce SAPALM, the first asynchronous parallel optimization method that provably converges for all nonconvex, nonsmooth problems of the form (1). SAPALM is a a block coordinate stochastic proximal-gradient method that generalizes the deterministic PALM method of [5, 3]. When applied to problem (1), we prove that SAPALM matches the best, known rates of convergence, due to [8] in the case where each $r_j$ is convex and $m = 1$: that is, asynchrony carries no theoretical penalty for convergence speed. We test SAPALM on a few example problems and compare to a synchronous implementation, showing a linear speedup.

**Notation.**  Let $m \in \mathbb{N}$ denote the number of coordinate blocks. We let $\mathcal{H} = \mathcal{H}_1 \times \ldots \times \mathcal{H}_m$. For every $x \in \mathcal{H}$, each partial gradient $\nabla_j f(x_1, \ldots, x_{j-1}, \cdot, x_{j+1}, \ldots, x_m) : \mathcal{H}_j \to \mathcal{H}_j$ is $L_j$-Lipschitz continuous; we let $\underline{L} = \min_j\{L_j\} \leq \max_j\{L_j\} = \overline{L}$. The number $\tau \in \mathbb{N}$ is the maximum allowable delay. Define the aggregate regularizer $r : \mathcal{H} \to (-\infty, \infty]$ as $r(x) = \sum_{j=1}^{m} r_j(x_j)$. For each $j \in \{1, \ldots, m\}$, $y \in \mathcal{H}_j$, and $\gamma > 0$, define the proximal operator

$$\mathbf{prox}_{\gamma r_j}(y) := \underset{x_j \in \mathcal{H}_j}{\mathrm{argmin}} \left\{ r_j(x_j) + \frac{1}{2\gamma}\|x_j - y\|^2 \right\}$$

For convex $r_j$, $\mathbf{prox}_{\gamma r_j}(y)$ is uniquely defined, but for nonconvex problems, it is, in general, a set. We make the mild assumption that for all $y \in \mathcal{H}_j$, we have $\mathbf{prox}_{\gamma r_j}(y) \neq \emptyset$. A slight technicality arises from our ability to choose among multiple elements of $\mathbf{prox}_{\gamma r_j}(y)$, especially in light of the stochastic nature of SAPALM. Thus, for all $y, j$ and $\gamma > 0$, we fix an element

$$\zeta_j(y, \gamma) \in \mathbf{prox}_{\gamma r_j}(y). \tag{2}$$

By [17, Exercise 14.38], we can assume that $\zeta_j$ is measurable, which enables us to reason with expectations wherever they involve $\zeta_j$. As shorthand, we use $\mathbf{prox}_{\gamma r_j}(y)$ to denote the (unique) choice $\zeta_j(y, \gamma)$. For any random variable or vector $X$, we let $\mathbb{E}_k[X] = \mathbb{E}\left[X \mid x^k, \ldots, x^0, \nu^k, \ldots, \nu^0\right]$ denote the conditional expectation of $X$ with respect to the sigma algebra generated by the history of SAPALM.

## 2  Algorithm Description

Algorithm 1 displays the SAPALM method.

We highlight a few features of the algorithm which we discuss in more detail below.

---

**Algorithm 1** SAPALM [Local view]

---

**Input:** $x \in \mathcal{H}$
1: All processors in parallel do
2: **loop**
3:     Randomly select a coordinate block $j \in \{1, \ldots, m\}$
4:     Read $x$ from shared memory
5:     Compute $g = \nabla_j f(x) + \nu_j$
6:     Choose stepsize $\gamma_j \in \mathbb{R}_{++}$                          ▷ According to Assumption 3
7:     $x_j \leftarrow \mathbf{prox}_{\gamma_j r_j}(x_j - \gamma_j g)$              ▷ According to (2)

---

- *Inconsistent iterates.* Other processors may write updates to $x$ in the time required to read $x$ from memory.

- *Coordinate blocks.* When the coordinate blocks $x_j$ are low dimensional, it reduces the likelihood that one update will be immediately erased by another, simultaneous update.

- *Noise.* The noise $\nu \in \mathcal{H}$ is a random variable that we use to model injected noise. It can be set to 0, or chosen to accelerate each iteration, or to avoid saddle points.

Algorithm 1 has an equivalent (mathematical) description which we present in Algorithm 2, using an iteration counter $k$ which is incremented each time a processor completes an update. This iteration counter is *not* required by the processors themselves to compute the updates.

In Algorithm 1, a processor might not have access to the shared-memory's global state, $x^k$, at iteration $k$. Rather, because all processors can continuously update the global state while other processors are reading, local processors might only read the *inconsistently delayed iterate* $x^{k-d_k} = (x_1^{k-d_{k,1}}, \ldots, x_m^{k-d_{k,m}})$, where the delays $d_k$ are integers less than $\tau$, and $x^l = x^0$ when $l < 0$.

---

**Algorithm 2** SAPALM [Global view]

---

**Input:** $x^0 \in \mathcal{H}$
1: **for** $k \in \mathbb{N}$ **do**
2:     Randomly select a coordinate block $j_k \in \{1, \ldots, m\}$
3:     Read $x^{k-d_k} = (x_1^{k-d_{k,1}}, \ldots, x_m^{k-d_{k,m}})$ from shared memory
4:     Compute $g^k = \nabla_{j_k} f(x^{k-d_k}) + \nu_{j_k}^k$
5:     Choose stepsize $\gamma_{j_k}^k \in \mathbb{R}_{++}$                          ▷ According to Assumption 3
6:     **for** $j = 1, \ldots, m$ **do**
7:         **if** $j = j_k$ **then**
8:             $x_{j_k}^{k+1} \leftarrow \mathbf{prox}_{\gamma_{j_k}^k r_{j_k}}(x_{j_k}^k - \gamma_{j_k}^k g^k)$        ▷ According to (2)
9:         **else**
10:            $x_j^{k+1} \leftarrow x_j^k$

---

## 2.1 Assumptions on the Delay, Independence, Variance, and Stepsizes

**Assumption 1** (Bounded Delay). *There exists some $\tau \in \mathbb{N}$ such that, for all $k \in \mathbb{N}$, the sequence of coordinate delays lie within $d_k \in \{0, \ldots, \tau\}^m$.*

**Assumption 2** (Independence). *The indices $\{j_k\}_{k \in \mathbb{N}}$ are uniformly distributed and collectively IID. They are independent from the history of the algorithm $x^k, \ldots, x^0, \nu^k, \ldots, \nu^0$ for all $k \in \mathbb{N}$.*

We employ two possible restrictions on the noise sequence $\nu^k$ and the sequence of allowable stepsizes $\gamma_j^k$, all of which lead to different convergence rates:

**Assumption 3** (Noise Regimes and Stepsizes). *Let $\sigma_k^2 := \mathbb{E}_k\left[\|\nu_k\|^2\right]$ denote the expected squared norm of the noise, and let $a \in (1, \infty)$. Assume that $\mathbb{E}_k\left[\nu^k\right] = 0$ and that there is a sequence of weights $\{c_k\}_{k \in \mathbb{N}} \subseteq [1, \infty)$ such that*

$$(\forall k \in \mathbb{N}), (\forall j \in \{1, \ldots, m\}) \qquad \gamma_j^k := \frac{1}{ac_k(L_j + 2L\tau m^{-1/2})}.$$

*which we choose using the following two rules, both of which depend on the growth of $\sigma_k$:*

**Summable.**  $\sum_{k=0}^{\infty} \sigma_k^2 < \infty \qquad \implies c_k \equiv 1;$

**$\alpha$-Diminishing.** $(\alpha \in (0,1)) \quad \sigma_k^2 = O((k+1)^{-\alpha}) \quad \implies c_k = \Theta((k+1)^{(1-\alpha)}).$

More noise, measured by $\sigma_k$, results in worse convergence rates and stricter requirements regarding which stepsizes can be chosen. We provide two stepsize choices which, depending on the noise regime, interpolate between $\Theta(1)$ and $\Theta(k^{1-\alpha})$ for any $\alpha \in (0,1)$. Larger stepsizes lead to convergence rates of order $O(k^{-1})$, while smaller ones lead to order $O(k^{-\alpha})$.

## 2.2  Algorithm Features

**Inconsistent Asynchronous Reading.**  SAPALM allows asynchronous access patterns. A processor may, at any time, and without notifying other processors:

1. **Read.** While other processors are writing to shared-memory, read the possibly out-of-sync, delayed coordinates $x_1^{k-d_{k,1}}, \ldots, x_m^{k-d_{k,m}}$.

2. **Compute.** Locally, compute the partial gradient $\nabla_{j_k} f(x_1^{k-d_{k,1}}, \ldots, x_m^{k-d_{k,m}})$.

3. **Write.** After computing the gradient, replace the $j_k$th coordinate with

$$x_{j_k}^{k+1} \in \underset{y}{\operatorname{argmin}}\, r_{j_k}(y) + \langle \nabla_{j_k} f(x^{k-d_k}) + \nu_{j_k}^k, y - x_{j_k}^k \rangle + \frac{1}{2\gamma_{j_k}^k}\|y - x_{j_k}^k\|^2.$$

Uncoordinated access eliminates waiting time for processors, which speeds up computation. The processors are blissfully ignorant of any conflict between their actions, and the paradoxes these conflicts entail: for example, the states $x_1^{k-d_{k,1}}, \ldots, x_m^{k-d_{k,m}}$ need never have simultaneously existed in memory. Although we write the method with a global counter $k$, the asynchronous processors need not be aware of it; and the requirement that the delays $d_k$ remain bounded by $\tau$ does not demand coordination, but rather serves only to define $\tau$.

**What Does the Noise Model Capture?**  SAPALM is the first asynchronous PALM algorithm to allow and analyze noisy updates. The stochastic noise, $\nu^k$, captures three phenomena:

1. **Computational Error.** Noise due to random computational error.
2. **Avoiding Saddles.** Noise deliberately injected for the purpose of avoiding saddles, as in [7].
3. **Stochastic Gradients.** Noise due to stochastic approximations of delayed gradients.

Of course, the noise model also captures any combination of the above phenomena. The last one is, perhaps, the most interesting: it allows us to prove convergence for a stochastic- or minibatch-gradient version of APALM, rather than requiring processors to compute a full (delayed) gradient. Stochastic gradients can be computed faster than their batch counterparts, allowing more frequent updates.

## 2.3  SAPALM as an Asynchronous Block Mini-Batch Stochastic Proximal-Gradient Method

In Algorithm 1, any stochastic estimator $\nabla f(x^{k-d_k}; \xi)$ of the gradient may be used, as long as $\mathbb{E}_k\left[\nabla f(x^{k-d_k}; \xi)\right] = \nabla f(x^{k-d_k})$, and $\mathbb{E}_k\left[\|\nabla f(x^{k-d_k}; \xi) - \nabla f(x^{k-d_k})\|^2\right] \le \sigma^2$. In particular, if Problem 1 takes the form

$$\underset{x \in \mathcal{H}}{\operatorname{minimize}}\, \mathbb{E}_\xi\left[f(x_1, \ldots, x_m; \xi)\right] + \frac{1}{m}\sum_{j=1}^{m} r_j(x_j),$$

then, in Algorithm 2, the stochastic mini-batch estimator $g^k = m_k^{-1}\sum_{i=1}^{m_k} \nabla f(x^{k-d_k}; \xi_i)$, where $\xi_i$ are IID, may be used in place of $\nabla f(x^{k-d_k}) + \nu^k$. A quick calculation shows that $\mathbb{E}_k\left[\|g^k - \nabla f(x^{k-d_k})\|^2\right] = O(m_k^{-1})$. Thus, any increasing batch size $m_k = \Omega((k+1)^{-\alpha})$, with $\alpha \in (0,1)$, conforms to Assumption 3.

When nonsmooth regularizers are present, all known convergence rate results for nonconvex stochastic gradient algorithms require the use of increasing, rather than fixed, minibatch sizes; see [8, 22] for analogous, synchronous algorithms.

# 3   Convergence Theorem

**Measuring Convergence for Nonconvex Problems.**   For nonconvex problems, it is standard to measure convergence (to a stationary point) by the *expected violation of stationarity*, which for us is the (deterministic) quantity:

$$S_k := \mathbb{E}\left[ \sum_{j=1}^{m} \left\| \frac{1}{\gamma_j^k}(w_j^k - x_j^k) + \nu_k \right\|^2 \right];$$

$$\text{where} \quad (\forall j \in \{1, \dots, m\}) \qquad w_j^k = \mathbf{prox}_{\gamma_j^k r_j}(x_j^k - \gamma_j^k(\nabla_j f(x^{k-d_k}) + \nu_j^k)). \tag{3}$$

A reduction to the case $r \equiv 0$ and $d_k = 0$ reveals that $w_j^k - x_j^k + \gamma_j^k \nu_j^k = -\gamma_j^k \nabla_j f(x^k)$ and, hence, $S_k = \mathbb{E}\left[ \|\nabla f(x^k)\|^2 \right]$. More generally, $w_j^k - r_j^k + \gamma_j^k \nu_j^k \in -\gamma_j^k(\partial_L r_j(w_j^k) + \nabla_j f(x^{k-d_k}))$ where $\partial_L r_j$ is the *limiting subdifferential* of $r_j$ [17] which, if $r_j$ is convex, reduces to the standard convex subdifferential familiar from [14]. A messy but straightforward calculation shows that our convergence rates for $S_k$ can be converted to convergence rates for elements of $\partial_L r(w^k) + \nabla f(w^k)$.

We present our main convergence theorem now and defer the proof to Section 4.

**Theorem 1** (SAPALM Convergence Rates). *Let $\{x^k\}_{k \in \mathbb{N}} \subseteq \mathcal{H}$ be the SAPALM sequence created by Algorithm 2. Then, under Assumption 3 the following convergence rates hold: for all $T \in \mathbb{N}$, if $\{\nu^k\}_{k \in \mathbb{N}}$ is*

1. *Summable*, then

$$\min_{k=0,\dots,T} S_k \le \mathbb{E}_{k \sim P_T}\left[ S_k \right] = O\left( \frac{m(\overline{L} + 2L\tau m^{-1/2})}{T+1} \right);$$

2. *$\alpha$-Diminishing*, then

$$\min_{k=0,\dots,T} S_k \le \mathbb{E}_{k \sim P_T}\left[ S_k \right] = O\left( \frac{m(\overline{L} + 2L\tau m^{-1/2}) + m\log(T+1)}{(T+1)^{-\alpha}} \right);$$

*where, for all $T \in \mathbb{N}$, $P_T$ is the distribution $\{0, \dots, T\}$ such that $P_T(X = k) \propto c_k^{-1}$.*

**Effects of Delay and Linear Speedups.**   The $m^{-1/2}$ term in the convergence rates presented in Theorem 1 prevents the delay $\tau$ from dominating our rates of convergence. In particular, as long as $\tau = O(\sqrt{m})$, the convergence rate in the synchronous ($\tau = 0$) and asynchronous cases are within a small constant factor of each other. In that case, because the work per iteration in the synchronous and asynchronous versions of SAPALM is the same, we expect a linear speedup: SAPALM with $p$ processors will converge nearly $p$ times faster than PALM, since the iteration counter will be updated $p$ times as often. As a rule of thumb, $\tau$ is roughly proportional to the number of processors. Hence we can achieve a linear speedup on as many as $O(\sqrt{m})$ processors.

## 3.1   The Asynchronous Stochastic Block Gradient Method

If the regularizer $r$ is identically zero, then the noise $\nu^k$ need not vanish in the limit. The following theorem guarantees convergence of asynchronous stochastic block gradient descent with a constant minibatch size. See the supplemental material for a proof.

**Theorem 2** (SAPALM Convergence Rates ($r \equiv 0$)). *Let $\{x^k\}_{k \in \mathbb{N}} \subseteq \mathcal{H}$ be the SAPALM sequence created by Algorithm 2 in the case that $r \equiv 0$. If, for all $k \in \mathbb{N}$, $\{\mathbb{E}_k\left[ \|\nu^k\|^2 \right]\}_{k \in \mathbb{N}}$ is bounded (not necessarily diminishing) and*

$$(\exists a \in (1, \infty)), (\forall k \in \mathbb{N}), (\forall j \in \{1, \dots, m\}) \qquad \gamma_j^k := \frac{1}{a\sqrt{k}(L_j + 2M\tau m^{-1/2})},$$

*then for all $T \in \mathbb{N}$, we have*

$$\min_{k=0,\dots,T} S_k \le \mathbb{E}_{k \sim P_T}\left[ S_k \right] = O\left( \frac{m(\overline{L} + 2L\tau m^{-1/2}) + m\log(T+1)}{\sqrt{T+1}} \right),$$

*where $P_T$ is the distribution $\{0, \dots, T\}$ such that $P_T(X = k) \propto k^{-1/2}$.*

# 4 Convergence Analysis

## 4.1 The Asynchronous Lyapunov Function

Key to the convergence of SAPALM is the following *Lyapunov function*, defined on $\mathcal{H}^{1+\tau}$, which aggregates not only the current state of the algorithm, as is common in synchronous algorithms, but also the history of the algorithm over the delayed time steps: $(\forall x(0), x(1), \ldots, x(\tau) \in \mathcal{H})$

$$\Phi(x(0), x(1), \ldots, x(\tau)) = f(x(0)) + r(x(0)) + \frac{L}{2\sqrt{m}} \sum_{h=1}^{\tau} (\tau - h + 1)\|x(h) - x(h-1)\|^2.$$

This Lyapunov function appears in our convergence analysis through the following inequality, which is proved in the supplemental material.

**Lemma 1** (Lyapunov Function Supermartingale Inequality). *For all $k \in \mathbb{N}$, let $z^k = (x^k, \ldots, x^{k-\tau}) \in \mathcal{H}^{1+\tau}$. Then for all $\epsilon > 0$, we have*

$$\mathbb{E}_k \left[ \Phi(z^{k+1}) \right] \leq \Phi(z^k) - \frac{1}{2m} \sum_{j=1}^{m} \left( \frac{1}{\gamma_j^k} - (1 + \epsilon) \left( L_j + \frac{2L\tau}{m^{1/2}} \right) \right) \mathbb{E}_k \left[ \|w_j^k - x_j^k + \gamma_j^k \nu_j^k\|^2 \right]$$

$$+ \sum_{j=1}^{m} \frac{\gamma_j^k \left( 1 + \gamma_j^k (1 + \epsilon^{-1}) \left( L_j + 2L\tau m^{-1/2} \right) \right) \mathbb{E}_k \left[ \|\nu_j^k\|^2 \right]}{2m}$$

*where for all $j \in \{1, \ldots, m\}$, we have $w_j^k = \mathbf{prox}_{\gamma_j^k r_j}(x_j^k - \gamma_j^k (\nabla_j f(x^{k-d_k}) + \nu_j^k))$. In particular, for $\sigma_k = 0$, we can take $\epsilon = 0$ and assume the last line is zero.*

Notice that if $\sigma_k = \epsilon = 0$ and $\gamma_j^k$ is chosen as suggested in Algorithm 2, the (conditional) expected value of the Lyapunov function is strictly decreasing. If $\sigma_k$ is nonzero, the factor $\epsilon$ will be used in concert with the stepsize $\gamma_j^k$ to ensure that noise does not cause the algorithm to diverge.

## 4.2 Proof of Theorem 1

For either noise regime, we define, for all $k \in \mathbb{N}$ and $j \in \{1, \ldots, m\}$, the factor $\epsilon := 2^{-1}(a - 1)$. With the assumed choice of $\gamma_j^k$ and $\epsilon$, Lemma 1 implies that the expected Lyapunov function decreases, up to a summable residual: with $A_j^k := w_j^k - x_j^k + \gamma_j^k \nu_j^k$, we have

$$\mathbb{E} \left[ \Phi(z^{k+1}) \right] \leq \mathbb{E} \left[ \Phi(z^k) \right] - \mathbb{E} \left[ \frac{1}{2m} \sum_{j=1}^{m} \frac{1}{\gamma_j^k} \left( 1 - \frac{1+\epsilon}{ac_k} \right) \|A_j^k\|^2 \right]$$

$$+ \sum_{j=1}^{m} \frac{\gamma_j^k \left( 1 + \gamma_j^k (1 + \epsilon^{-1}) \left( L_j + 2L\tau m^{-1/2} \right) \right) \mathbb{E} \left[ \mathbb{E}_k \left[ \|\nu_j^k\|^2 \right] \right]}{2m}. \quad (4)$$

Two upper bounds follow from the the definition of $\gamma_j^k$, the lower bound $c_k \geq 1$, and the straightforward inequalities $(ac_k)^{-1}(\underline{L} + 2M\tau m^{-1/2})^{-1} \geq \gamma_j^k \geq (ac_k)^{-1}(\overline{L} + 2M\tau m^{-1/2})^{-1}$:

$$\frac{1}{c_k} S_k \leq \frac{1}{\frac{(1-(1+\epsilon)a^{-1})}{2ma(\overline{L}+2L\tau m^{-1/2})}} \mathbb{E} \left[ \frac{1}{2m} \sum_{j=1}^{m} \frac{1}{\gamma_j^k} \left( 1 - \frac{1+\epsilon}{ac_k} \right) \|A_j^k\|^2 \right]$$

and

$$\sum_{j=1}^{m} \frac{\gamma_j^k \left( 1 + \gamma_j^k (1 + \epsilon^{-1}) \left( L_j + 2L\tau m^{-1/2} \right) \right) \mathbb{E}_k \left[ \|\nu_j^k\|^2 \right]}{2m} \leq \frac{(1 + (ac_k)^{-1}(1 + \epsilon^{-1}))(\sigma_k^2/c_k)}{2a(\underline{L} + 2L\tau m^{-1/2})}.$$

Now rearrange (4), use $\mathbb{E} \left[ \Phi(z^{k+1}) \right] \geq \inf_{x \in \mathcal{H}} \{f(x) + r(x)\}$ and $\mathbb{E} \left[ \Phi(z^0) \right] = f(x^0) + r(x^0)$, and sum (4) over $k$ to get

$$\frac{1}{\sum_{k=0}^{T} c_k^{-1}} \sum_{k=0}^{T} \frac{1}{c_k} S_k \leq \frac{f(x^0) + r(x^0) - \inf_{x \in \mathcal{H}} \{f(x) + r(x)\} + \sum_{k=0}^{T} \frac{(1+(ac_k)^{-1}(1+\epsilon^{-1}))(\sigma_k^2/c_k)}{2a(\underline{L}+2L\tau m^{-1/2})}}{\frac{(1-(1+\epsilon)a^{-1})}{2ma(\overline{L}+2L\tau m^{-1/2})} \sum_{k=0}^{T} c_k^{-1}}.$$

The left hand side of this inequality is bounded from below by $\min_{k=0,\ldots,T} S_k$ and is precisely the term $\mathbb{E}_{k \sim P_T}[S_k]$. What remains to be shown is an upper bound on the right hand side, which we will now call $R_T$.

If the noise is summable, then $c_k \equiv 1$, so $\sum_{k=0}^{T} c_k^{-1} = (T+1)$ and $\sum_{k=0}^{T} \sigma_k^2/c_k < \infty$, which implies that $R_T = O(m(\overline{L} + 2L\tau m^{-1/2})(T+1)^{-1})$. If the noise is $\alpha$-diminishing, then $c_k = \Theta(k^{(1-\alpha)})$, so $\sum_{k=0}^{T} c_k^{-1} = \Theta((T+1)^\alpha)$ and, because $\sigma_k^2/c_k = O(k^{-1})$, there exists a $B > 0$ such that $\sum_{k=0}^{T} \sigma_k^2/c_k \leq \sum_{k=0}^{T} Bk^{-1} = O(\log(T+1))$, which implies that $R_T = O((m(\overline{L} + 2L\tau m^{-1/2}) + m\log(T+1))(T+1)^{-\alpha})$.

## 5    Numerical Experiments

In this section, we present numerical results to confirm that SAPALM delivers the expected performance gains over PALM. We confirm two properties: 1) SAPALM converges to values nearly as low as PALM given the same number of iterations, 2) SAPALM exhibits a near-linear speedup as the number of workers increases. All experiments use an Intel Xeon machine with 2 sockets and 10 cores per socket.

We use two different nonconvex matrix factorization problems to exhibit these properties, to which we apply two different SAPALM variants: one without noise, and one with stochastic gradient noise. For each of our examples, we generate a matrix $A \in \mathbb{R}^{n \times n}$ with iid standard normal entries, where $n = 2000$. Although SAPALM is intended for use on much larger problems, using a small problem size makes write conflicts more likely, and so serves as an ideal setting to understand how asynchrony affects convergence.

1. **Sparse PCA with Asynchronous Block Coordinate Updates.** We minimize

$$\operatorname*{argmin}_{X,Y} \frac{1}{2}||A - X^T Y||_F^2 + \lambda\|X\|_1 + \lambda\|Y\|_1, \tag{5}$$

where $X \in \mathbb{R}^{d \times n}$ and $Y \in \mathbb{R}^{d \times n}$ for some $d \in \mathbb{N}$. We solve this problem using SAPALM with no noise $\nu^k = 0$.

2. **Quadratically Regularized Firm Thresholding PCA with Asynchronous Stochastic Gradients.** We minimize

$$\operatorname*{argmin}_{X,Y} \frac{1}{2}||A - X^T Y||_F^2 + \lambda(\|X\|_{\text{Firm}} + \|Y\|_{\text{Firm}}) + \frac{\mu}{2}(\|X\|_F^2 + \|Y\|_F^2), \tag{6}$$

where $X \in \mathbb{R}^{d \times n}, Y \in \mathbb{R}^{d \times n}$, and $\|\cdot\|_{\text{Firm}}$ is the firm thresholding penalty proposed in [21]: a **nonconvex, nonsmooth** function whose proximal operator truncates small values to zero and preserves large values. We solve this problem using the stochastic gradient SAPALM variant from Section 2.3.

In both experiments $X$ and $Y$ are treated as coordinate blocks. Notice that for this problem, the SAPALM update decouples over the entries of each coordinate block. Each worker updates its coordinate block (say, $X$) by cycling through the coordinates of $X$ and updating each in turn, restarting at a random coordinate after each cycle.

In Figures (1a) and (1c), we see objective function values plotted by iteration. By this metric, SAPALM performs as well as PALM, its single threaded variant; for the second problem, the curves for different thread counts all overlap. Note, in particular, that SAPALM does not diverge. But SAPALM can add additional workers to increment the iteration counter more quickly, as seen in Figure 1b, allowing SAPALM to outperform its single threaded variant.

We measure the speedup $S_k(p)$ of SAPALM by the (relative) time for $p$ workers to produce $k$ iterates

$$S_k(p) = \frac{T_k(p)}{T_k(1)}, \tag{7}$$

where $T_k(p)$ is the time to produce $k$ iterates using $p$ workers. Table 2 shows that SAPALM achieves near linear speedup for a range of variable sizes $d$. (Dashes — denote experiments not run.)

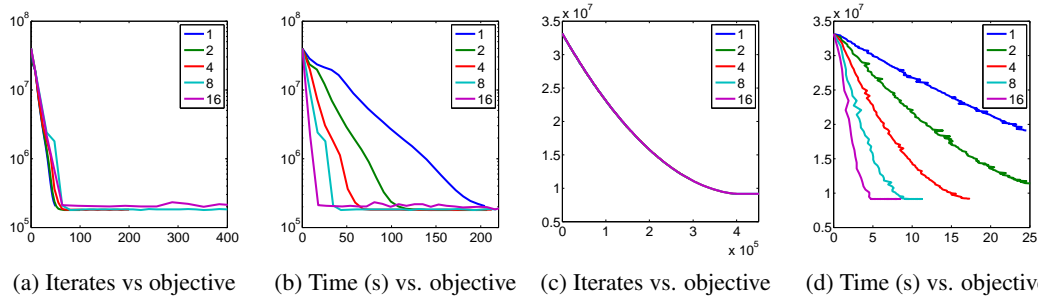

| (a) Iterates vs objective | (b) Time (s) vs. objective | (c) Iterates vs. objective | (d) Time (s) vs. objective |

Figure 1: Sparse PCA ((1a) and (1b)) and Firm Thresholding PCA ((1c) and (1d)) tests for $d = 10$.

| threads | d=10 | d=20 | d=100 |
|---|---|---|---|
| 1 | 65.9972 | 253.387 | 6144.9427 |
| 2 | 33.464 | 127.8973 | – |
| 4 | 17.5415 | 67.3267 | – |
| 8 | 9.2376 | 34.5614 | 833.5635 |
| 16 | 4.934 | 17.4362 | 416.8038 |

Table 1: Sparse PCA timing for 16 iterations by problem size and thread count.

| threads | d=10 | d=20 | d=100 |
|---|---|---|---|
| 1 | 1 | 1 | 1 |
| 2 | 1.9722 | 1.9812 | – |
| 4 | 3.7623 | 3.7635 | – |
| 8 | 7.1444 | 7.3315 | 7.3719 |
| 16 | 13.376 | 14.5322 | 14.743 |

Table 2: Sparse PCA speedup for 16 iterations by problem size and thread count.

Deviations from linearity can be attributed to a breakdown in the abstraction of a "shared memory" computer: as each worker modifies the "shared" variables $X$ and $Y$, some communication is required to maintain cache coherency across all cores and processors. In addition, Intel Xeon processors share L3 cache between all cores on the processor. All threads compete for the same L3 cache space, slowing down each iteration. For small $d$, write conflicts are more likely; for large $d$, communication to maintain cache coherency dominates.

## 6 Discussion

A few straightforward generalizations of our work are possible; we omit them to simplify notation.

**Removing the** $\log$ **factors.** The $\log$ factors in Theorem 1 can easily be removed by fixing a maximum number of iterations for which we plan to run SAPALM and adjusting the $c_k$ factors accordingly, as in [14, Equation (3.2.10)].

**Cluster points of** $\{x^k\}_{k\in\mathbb{N}}$**.** Using the strategy employed in [5], it's possible to show that all cluster points of $\{x^k\}_{k\in\mathbb{N}}$ are (almost surely) stationary points of $f + r$.

**Weakened Assumptions on Lipschitz Constants.** We can weaken our assumptions to allow $L_j$ to vary: we can assume $L_j(x_1, \ldots, x_{j-1}, \cdot, x_{j+1}, \ldots, x_m)$-Lipschitz continuity each partial gradient $\nabla_j f(x_1, \ldots, x_{j-1}, \cdot, x_{j+1}, \ldots, x_m) : \mathcal{H}_j \to \mathcal{H}_j$, for every $x \in \mathcal{H}$.

## 7 Conclusion

This paper presented SAPALM, the first asynchronous parallel optimization method that provably converges on a large class of nonconvex, nonsmooth problems. We provide a convergence theory for SAPALM, and show that with the parameters suggested by this theory, SAPALM achieves a near linear speedup over serial PALM. As a special case, we provide the first convergence rate for (synchronous or asynchronous) stochastic block proximal gradient methods for nonconvex regularizers. These results give specific guidance to ensure fast convergence of practical asynchronous methods on a large class of important, nonconvex optimization problems, and pave the way towards a deeper understanding of stability of these methods in the presence of noise.

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
