[Supplementary Material]

# Supplementary Material: The Sound of APALM Clapping: Faster Nonsmooth Nonconvex Optimization with Stochastic Asynchronous PALM

**Damek Davis and Madeleine Udell**
Cornell University
{dsd95,mru8}@cornell.edu

**Brent Edmunds**
University of California, Los Angeles
brent.edmunds@math.ucla.edu

## Abstract

This document contains a few proofs, which were omitted from our NIPS submission.

## 1 Proof of Lemma 1

**Lemma 1** (Lyapunov Function Supermartingale Inequality). *For all $k \in \mathbb{N}$, let $z^k = (x^k, \ldots, x^{k-\tau}) \in \mathcal{H}^{1+\tau}$. Then for all $\epsilon \in \mathbb{R}_{++}$, we have*

$$\mathbb{E}_k \left[ \Phi(z^{k+1}) \right] \leq \Phi(z^k) - \frac{1}{2m} \sum_{j=1}^m \left( \frac{1}{\gamma_j^k} - (1+\epsilon) \left( L_j + \frac{2L\tau}{m^{1/2}} \right) \right) \mathbb{E}_k \left[ \|w_j^k - x_j^k + \gamma_j^k \nu_j^k\|^2 \right]$$
$$+ \sum_{j=1}^m \frac{\gamma_j^k \left( 1 + \gamma_j^k (1+\epsilon^{-1}) \left( L_j + 2L\tau m^{-1/2} \right) \right) \mathbb{E}_k \left[ \|\nu_j^k\|^2 \right]}{2m}$$

*where for all $j \in \{1, \ldots, m\}$, we have $w_j^k \in \mathbf{prox}_{\gamma_j^k r_j}(x_j^k - \gamma_j^k(\nabla_j f(x^{k-d_k}) + \nu_j^k))$. In particular, for $\sigma_k = 0$, we can take $\epsilon = 0$ and assume the last line is zero.*

We first prove a descent property of the objective function—up to some residuals which are the result of asynchrony and noise:

**Lemma 2.** *For all $k \in \mathbb{N}$, we have*

$$\mathbb{E}_k \left[ f(x^{k+1}) + r(x^{k+1}) \right] \leq f(x^k) + r(x^k)$$
$$- \frac{1}{2m} \sum_{j=1}^m \left( \frac{1}{\gamma_j^k} - (1+\epsilon) L_j \right) \mathbb{E}_k \left[ \|w_j^k - x_j^k + \nu_j^k \nu_j^k\|^2 \right]$$
$$+ \sum_{j=1}^m \frac{\gamma_j^k}{2m} \left( 1 + (1+\epsilon^{-1}) L_j \gamma_j^k \right) \mathbb{E}_k \left[ \|\nu_j^k\|^2 \right]$$
$$+ \frac{1}{m} \mathbb{E}_k \left[ \langle \nabla f(x^k) - \nabla f(x^{k-d_k}), w^k - x^k \rangle \right].$$

*Proof.* The standard upper bound [2, Lemma 1.2.3] for functions with Lipschitz continuous gradients implies that

$$f(x_1, \ldots, w_j^k, \ldots, x_m^k) \leq f(x^k) + \langle w_j^k - x_j^k, \nabla f(x^k) \rangle + \frac{L_j}{2} \|w_j^k - x_j^k\|^2.$$
$$\leq f(x^k) + \langle w_j^k - x_j^k, \nabla f(x^k) \rangle$$
$$+ \frac{(1+\epsilon)L_j}{2} \|w_j^k - x_j^k + \gamma_j^k \nu_j^k\|^2 + \frac{(1+\epsilon^{-1})L_j}{2} \|\gamma_j^k \nu_j^k\|^2.$$

And the definition of $w_j^k$ as a proximal point implies that

$$r_j(w_j^k) \leq r(x_j^k) - \langle w_j^k - x_j^k + \gamma_j^k \nu_j^k, \nabla_j f(x^{k-d_k}) \rangle - \frac{1}{2\gamma_j^k} \|w_j^k - x_j^k + \gamma_j^k \nu_j^k\|^2 + \frac{1}{2\gamma_j^k} \|\gamma_j^k \nu_j^k\|^2.$$

Given these two inequalities and the identity $\mathbb{E}_k \left[\nu^k\right] = 0$, we have

$$\mathbb{E}_k \left[f(x^{k+1}) + r(x^{k+1})\right] \leq \frac{1}{m} \sum_{j=1}^m f(x_1^k, \ldots, w_j^k, \ldots, x_m^k) + \sum_{j=1}^m \left(\frac{1}{m} r_j(w_j^k) + \left(1 - \frac{1}{m}\right) r_j(x_j^k)\right)$$
$$\leq f(x^k) + r(x^k) - \frac{1}{2m} \sum_{j=1}^m \left(\frac{1}{\gamma_j^k} - (1+\epsilon)L_j\right) \mathbb{E}_k \left[\|w_j^k - x_j^k + \nu_j^k \nu_j^k\|^2\right]$$
$$+ \sum_{j=1}^m \frac{\gamma_j^k}{2m} \left(1 + (1+\epsilon^{-1})L_j \gamma_j^k\right) \mathbb{E}_k \left[\|\nu_j^k\|^2\right]$$
$$+ \frac{1}{m} \mathbb{E}_k \left[\langle \nabla f(x^k) - \nabla f(x^{k-d_k}), w^k - x^k \rangle\right].$$

$\square$

The residual due to asynchrony can be conveniently placed inside a sum that alternates up to a small noise residual:

**Lemma 3.** *For all $k \in \mathbb{N}$ and any $\epsilon \in \mathbb{R}_+^m$, we have*

$$\frac{L}{2\sqrt{m}} \sum_{h=(k+1)-\tau+1}^{k+1} ((k+1) - h + 1) \mathbb{E}_k \left[\|x^h - x^{h-1}\|^2\right]$$
$$\leq \frac{L}{2\sqrt{m}} \sum_{h=k-\tau+1}^k (k - h + 1) \|x^h - x^{h-1}\|^2 - \frac{1}{m} \mathbb{E}_k \left[\langle \nabla f(x^k) - \nabla f(x^{k-d_k}), w^k - x^k \rangle\right]$$
$$+ \frac{(1+\epsilon)2L\tau}{2m^{3/2}} \mathbb{E}_k \left[\|w^k - x^k + \gamma_j^k \nu_j^k\|^2\right] + \sum_{j=1}^m \frac{(1+\epsilon^{-1})2L\tau \mathbb{E}_k \left[\|\gamma_j^k \nu_j^k\|^2\right]}{2m^{3/2}}.$$

*Proof.* The asynchronous term splits into the sum of two alternating terms and a third easily handled term:[1] for all $C > 0$, we have

$$\frac{1}{m}\mathbb{E}_k\left[\langle\nabla f(x^k) - \nabla f(x^{k-d_k}), w^k - x^k\rangle\right]$$

$$\leq \frac{1}{m}\mathbb{E}_k\left[L\|x^k - x^{k-d_k}\|\|w^k - x^k\|\right]$$

$$\leq \mathbb{E}_k\left[\frac{L}{2\sqrt{m}\tau}\|x^k - x^{k-d_k}\|^2 + \frac{L\tau}{2m^{3/2}}\|w^k - x^k\|^2\right]$$

$$\leq \mathbb{E}_k\left[\frac{L}{2\tau\sqrt{m}}\sum_{j=1}^m d_{k,j}\sum_{h=k-d_{k,j}+1}^k\|x_j^h - x_j^{h-1}\|^2 + \frac{L\tau}{2m^{3/2}}\|w^k - x^k\|^2\right] \qquad \text{(by Jensen's inequality)}$$

$$\leq \mathbb{E}_k\left[\frac{L}{2\sqrt{m}}\sum_{j=1}^m\sum_{h=k-\tau+1}^k\|x_j^h - x_j^{h-1}\|^2 + \frac{L\tau}{2m^{3/2}}\|w^k - x^k\|^2\right]$$

$$= \mathbb{E}_k\left[\left(\frac{L}{2\sqrt{m}}\sum_{h=k-\tau+1}^k(h-k+\tau)\|x^h - x^{h-1}\|^2 - \frac{L}{2\sqrt{m}}\sum_{h=k-\tau+2}^{k+1}(h-(k+1)+\tau)\|x^h - x^{h-1}\|^2\right)\right.$$

$$\left. + \frac{L\tau}{2\sqrt{m}}\|x^{k+1} - x^k\|^2 + \frac{L\tau}{2m^{3/2}}\|w^k - x^k\|^2\right].$$

The proof is completed by noticing that $\mathbb{E}_k\left[\|x^{k+1} - x^k\|^2\right] = m^{-1}\mathbb{E}_k\left[\|w^k - x^k\|^2\right]$, combining the two terms on the last line, and using the following inequality:

$$\|w^k - x^k\|^2 \leq \sum_{j=1}^m(1+\epsilon)\|w_j^k - x_k^k + \gamma_j^k\nu_j^k\|^2 + \sum_{j=1}^m(1+\epsilon^{-1})\|\gamma_j^k\nu_j^k\|^2.$$

$\square$

Summing up the bounds in the Lemmas, we obtain the claimed decrease in the Lyapunov function.

## 2 Relaxed Assumptions on the Variance When $r \equiv 0$

It's easy to modify the Lyapunov function in the case that $r \equiv 0$ to the following form:

**Lemma 4** (Lyapunov Function Supermartingale Inequality). *For all $k \in \mathbb{N}$, let $z^k = (x^k, \ldots, x^{k-\tau}) \in \mathcal{H}^{1+\tau}$. Then for all $\epsilon \in \mathbb{R}_{++}$, we have*

$$\mathbb{E}_k\left[\Phi(z^{k+1})\right] \leq \Phi(z^k) - \frac{1}{2m}\sum_{j=1}^m\gamma_j^k\left(2 - (1+\epsilon)\left(L_j + \frac{2L\tau}{m^{1/2}}\right)\gamma_j^k\right)\|\nabla_j f(x^{k-d_k})\|^2$$

$$+ \sum_{j=1}^m\frac{(\gamma_j^k)^2(1+\epsilon^{-1})\left(L_j + 2L\tau m^{-1/2}\right)\mathbb{E}_k\left[\|\nu_j^k\|^2\right]}{2m}.$$

*In particular, for $\sigma_k = 0$, we can take $\epsilon = 0$ and assume the last line is zero.*

Key to this inequality is that, at each iteration, the noise variance is multiplied by $(\gamma_j^k)^2$, rather than by $\gamma_j^k$. Following the proof of Theorem 1 yields the following theorem in the case that $r \equiv 0$:

**Theorem 1** (SAPALM Convergence Rates ($r \equiv 0$)). *Let $\{x^k\}_{k\in\mathbb{N}} \subseteq \mathcal{H}$ be the SAPALM sequence created by Algorithm 1 in the case that $r \equiv 0$. If, for all $k \in \mathbb{N}$, $\{\mathbb{E}_k\left[\|\nu^k\|^2\right]\}_{k\in\mathbb{N}}$ is bounded (not necessarily diminishing), and*

$$\left(\exists a \in (1,\infty)\right), \left(\forall k \in \mathbb{N}\right), \left(\forall j \in \{1,\ldots,m\}\right) \qquad \gamma_j^k := \frac{1}{a\sqrt{k}(L_j + 2M\tau m^{-1/2})},$$

*then for all $T \in \mathbb{N}$, we have*

$$\min_{k=0,\dots,T} S_k \leq \mathbb{E}_{k \sim P_T}[S_k] = O\left(\frac{m(\overline{L} + 2L\tau m^{-1/2}) + m\log(T+1)}{\sqrt{T+1}}\right),$$

*where $P_T$ is the distribution $\{0, \dots, T\}$ such that $P_T(X = k) \propto k^{-1/2}$.*

Now for the decrease of the Lyapunov function:

*Proof of Lemma 4.* The standard upper bound [2, Lemma 1.2.3] for functions with Lipschitz continuous gradients implies that

$$f(x_1, \dots, w_j^k, \dots, x_m^k) = f(x^k) + \langle w_j^k - x_j^k, \nabla_j f(x^k)\rangle + \frac{L_j}{2}\|w_j^k - x_j^k\|^2.$$

$$\leq f(x^k) + \langle w_j^k - x_j^k, \nabla_j f(x^k)\rangle$$

$$+ \frac{(1+\epsilon)L_j}{2}\|w_j^k - x_j^k + \gamma_j^k \nu_j^k\|^2 + \frac{(1+\epsilon^{-1})L_j}{2}\|\gamma_j^k \nu_j^k\|^2.$$

The inner product term can be split into two further pieces

$$\mathbb{E}_k\left[\langle w_j^k - x_j^k, \nabla_j f(x^k)\rangle\right] = \mathbb{E}_k\left[\langle w_j^k - x_j^k + \gamma_j^k \nu_j^k, \nabla_j f(x^{k-d_k})\rangle\right] + \mathbb{E}_k\left[\langle w_j^k - x_j^k, \nabla_j f(x^k) - \nabla_j f(x^{k-d_k})\rangle\right],$$

where we've use the equality $\mathbb{E}_k[\nu_k] = 0$. Thus, owing to the equality $w_j^k - x_j^k + \gamma_j^k \nu_j^k = -\gamma_j^k \nabla_j f(x^{k-d_k})$, we have

$$\mathbb{E}_k\left[f(x^{k+1})\right] \leq \frac{1}{m}\sum_{j=1}^m f(x_1^k, \dots, w_j^k, \dots, x_m^k)$$

$$\leq f(x^k) - \sum_{j=1}^m \frac{\gamma_j^k(2 - (1+\epsilon)L_j\gamma_j^k)}{2m}\|\nabla_j f(x^{k-d_k})\|^2$$

$$+ \sum_{j=1}^m \frac{(1+\epsilon^{-1})L_j}{2m}\|\gamma_j^k \nu_j^k\|^2.$$

$$+ \frac{1}{m}\mathbb{E}_k\left[\langle \nabla f(x^k) - \nabla f(x^{k-d_k}), w^k - x^k\rangle\right].$$

The proof finished by combining this inequality with the inequality in Lemma 3.

$\square$

## Footnotes

[1] we use the same bound presented in [1, Theorem 4.1], but we reproduce it for completeness.