[Reviews · NeurIPS 2016]

Reviewer 1

Summary

The paper presents a block coordinate optimization algorithm using asynchronous parallel processing. Specifically, the proposed method is applicable to nonconvex nonsmooth problems of cost functions obeying a specific structure (Eq 1). The algorithm is easy to implement and is accompanied with a convergence analysis. The proposed algorithm achieves near linear speedup over similar algorithms working serially. The claims are supported by empirical results on two matrix factorization problems.

Qualitative Assessment

The presented algorithm and its accompanied guaranteed convergence are indeed very interesting. Nevertheless, there are a few places in the paper that are not quite clear to me: 1. Is the algorithm guaranteed to converge to a "local minimum"? The convergence analysis in Section 3 relies on the definition in Eq(3). The latter merely guarantees states convergence to a "stationary point". However, throughout the paper, authors mention that the addition of the noise v_k helps the algorithm escape from saddle points (Line 116, Item 2). It seems there is an argument by the authors about reaching a local minimum, which the analysis (as currently presented) does not provide. Please clarify. 2. Nonconvex Regularization: The authors state that the proposed algorithm/analysis applies to general nonconvex regularizers. The latter may give rise to non-unique proximal operators (Line 63). Given that handling general nonconvexity is stressed as a key property of the algorithm, it would be nice if handling such non-unique scenarios were explained in greater details. Unfortunately, it seems the nonconvex experiment (Firm Norm) also limited to the special case when the proximal operator can be defined uniquely (despite nonconvexity). In general, however, how should the algorithm choose among non-unique answers? Does a random choice uniformly across the set of proximal operators work? Or it may cause a problem if the randomly chosen elements between two successive iterates happen to be far from each other? 3. It is not stated what different colors in Figure 1 represent neither in the caption of Figure 1, nor in the main text). I suppose these are the number of threads, but it should be mentioned explicitly.

Confidence in this Review

1-Less confident (might not have understood significant parts)


Reviewer 2

Summary

This paper introduces a noisy asynchronous block coordinate descent method for solving nonconvex, nonsmooth optimization problems. The main contribution is its prove of convergence for a large class of optimization problems.

Qualitative Assessment

The algorithm proposed in this paper for implementing noisy asynchronous block coordinate descent is something that comes to the mind immediately if someone is to implement such an algorithm. The main contribution of this paper is providing proofs that such a simple algorithm actually works. It proves that the algorithm matches the best known rates of convergence on this problem class. The paper is well-written and it is very easy to follow the arguments of the paper. The authors start with a short introduction of known results and subsequently explain their algorithm and their main convergence theorem. Through experiments on two different non convex matrix factorization problems, they showed that the algorithm attain linear speed up on the number of cpu threads. Overall the paper is a valuable peace of work and its theoretical results can be of benefit for practitioner of large machine learning systems.

Confidence in this Review

2-Confident (read it all; understood it all reasonably well)


Reviewer 3

Summary

The paper proposes an asynchronous proximal gradient descent algorithm and provides a proof of its guaranteed convergence to a local optimal solution. The main contribution is the finding that the prox function for nonconvex problems defines a set which in turn induces a measurable space and to interpret the unchecked asynchronous updates of variables as a source of noise when computing the gradient. In doing so, the distributed computation can be seen as an instance of stochastically altered gradient descent algorithm. The authors report convergence behavior under two different noise regimes resulting in constant and decreasing step sizes respectively.

Qualitative Assessment

I find the approach rather interesting, especially the broad and general definition of the problem makes the approach applicable to a wide range of problems. However, I was surprised by the absence of any reference to the seminal Robbins/Munro paper and also to the recent developments in stochastic gradient descent based sampling (see below). The authors do local gradient descent updates of coordinate blocks by computing partial gradients and adding noise in each asynchronous step. I was wondering, how this relates to the "usual" stochastic gradient descent update, i.e., given that the locally computed partial gradient will be based on delayed (noisy) variable states, a sequence of these noisy partial gradients would converge to the true partial gradient as well. Further, recent SGD based sampling has shown that adding noise to the variable states obtained by noisy gradient updates (as the authors do as well) provides good samples of the distribution underlying the optimal variable setting also in a non-convex setting. That being said, the work handed in remains valid, but it would have been interesting to compare the proposed approach to well established stochastic gradient methods. The overall procedure is laid out well and comprehensible. The chosen examples in the experiments section are well suited to demonstrate the scalability benefits of the algorithm. Minor to that, I have a few remarks on style and the overall rationale: - line 60: "each" is unnecessary here when m = 1 - line 69: k is not yet defined as are g, and \nu - line 106: the notation using y is probably wrong, shouldn't it read argmin_{x_{j_k}^k} r_{j_k} (x_{j_k}^k) + ... ? - Algorithms 1 and 2 lack a break condition and output - Table 1: I assume the timing is in seconds? Literature: Robbins, H., & Monro, S. (1951). A stochastic approximation method. The Annals of Mathematical Statistics, 400–407. Welling, M., & Teh, Y.-W. (2011). Bayesian learning via stochastic gradient Langevin dynamics (pp. 681–688). Presented at the Proceedings of the 28th International Conference on Machine Learning.

Confidence in this Review

2-Confident (read it all; understood it all reasonably well)


Reviewer 4

Summary

In this paper, the authors proposed NAPALM for solving nonconvex, nonsmooth optimization problems. And they claim that NAPALM is the first asynchronous parallel optimization method that provably converges on a large class of nonconvex,nonsmooth problems. Moreover, the authors prove iteration complexity that NAPALM and demonstrate state-of-the-art performance on several matrix factorization problems.

Qualitative Assessment

(1) This paper combines the two optimization techniques called "Asynchronous parallel" and PALM or BGCD for nonconvex nonsmooth optimization problems. In fact, the main proof techniques are standard. Hence, I do not find the results very exciting. (2) The authors claim that their method NAPALM mathches the best known convergence rates. In fact, this optimal rate only holds for the summable error case. The authors should make it clear. (3) This paper also covers the asynchronous stochastic block gradient descent.However, the convergence analysis holds only for the nonconvex, smooth optimization.

Confidence in this Review

2-Confident (read it all; understood it all reasonably well)


Reviewer 5

Summary

The authors consider asynchronous coordinate descent with noise and possibly nonsmooth regularizer for nonconvex optimization problems, and provide the proof for the convergence rate.

Qualitative Assessment

The paper proves the convergence rate for asynchronous coordinate descent algorithm with nonsmooth regularizer and noise on nonconvex problems. The main contribution is a generalization of the bounds of Ji Liu and Stephen J. Wright's 2014 paper on asynchronous coordinate descent (this paper cited their 2013 work, but to me the 2014 work is more relevant) to nonconvex optimization problems with some noise on the gradients. This topic is very interesting since there are many applications for SCD under nonconvex setting. I have a few questions about this paper: 1. In line 77, the paper says another update may overwrite previous updates, but from algorithm 2 it seems that all updates are effective. 2. In line 135, the paper says the expected violation of stationarity is the standard measure. Is there any reference for that? 3. From theorem 2 it seems that the theorem only talks about the convergence of increasing batch size SGD. Do you think it is easy to derive the convergence rate of constant batch size SGD like the reference [8] using the same technique in this paper?

Confidence in this Review

2-Confident (read it all; understood it all reasonably well)


Reviewer 6

Summary

This paper proposes a noisy asynchronous PALM algorithm to solve general nonconvex nonsmooth optimization problems. The algorithm is actually a block coordinate stochastic proximal gradient method. The paper gives detailed convergence analysis and can get linear speedup in experiments.

Qualitative Assessment

This paper is well written and easy to read. The main contribution of this paper is adding the noise to the stochastic coordinate gradient in the asynchronous PALM framework, which is also the main difference compared with the previous work [5]. But I think [5] gives more rigorous theoretical analysis and insights. The authors get different convergence rate under different noise, this is a good point.

Confidence in this Review

3-Expert (read the paper in detail, know the area, quite certain of my opinion)